**Data Availability Statement:** All relevant data are within the paper and its Supporting information files.

# Reducing intraventricular hemorrhage following the implementation of a prevention bundle for neonatal hypothermia

**Wei-Tse Chiu**[1,2], **Yi-Hsuan Lu**[2,3☯], **Yin-Ting Chen**[4,5☯], **Yin Ling Tan**[2,6], **Yi-Chieh Lin**[2,7], **Yu-Lien Chen**[8], **Hung-Chieh Chou**[2], **Chien-Yi Chen**[2], **Ting-An Yen**[2], **Po-Nien Tsao**[2,9]*

**1** Department of Pediatrics, National Taiwan University Hospital, Yun-Lin Branch, Yun-Lin, Taiwan, **2** Department of Pediatrics, National Taiwan University Children Hospital, Taipei, Taiwan, **3** Department of Pediatrics, National Taiwan University Hospital, Hsin-Chu Branch, Hsin-Chu, Taiwan, **4** Division of Neonatology, Department of Pediatrics, Children Hospital, China Medical University, Taichung, Taiwan, **5** Department of Medicine, School of Medicine, China Medical University, Taichung, Taiwan, **6** Department of Pediatrics, Fu Jen Catholic University Hospital, Taipei, Taiwan, **7** Taoyuan General Hospital, Ministry of Health and Welfare, Taoyuan, Taiwan, **8** Department of Nursery, National Taiwan University Hospital, Taipei, Taiwan, **9** Research Center for Developmental Biology & Regenerative Medicine, National Taiwan University, Taipei, Taiwan

☯ These authors contributed equally to this work.

* tsaopn@ntu.edu.tw

## Abstract

### Introduction

In very low birth weight (VLBW) infants, hypothermia immediately following birth is common even in countries rich in medical resources. The purpose of this study is to design a standard prevention bundle that decreases the rate of hypothermia among infants after birth and to investigate efficacy of the bundle and short-term outcomes for VLBW infants.

### Methods

This quality improvement project was conducted from February 2017 to July 2018 on all VLBW preterm infants admitted at a single referral level III neonatal intensive care unit. The infants were classified into the pre-intervention (February to September 2017) and post-intervention (October 2017 to July 2018) groups according to the time periods when they were recruited. During the pre-intervention period, we analyzed the primary causes of hypothermia, developed solutions corresponding to each cause, integrated all solutions into a prevention bundle, and applied the bundle during the post-intervention period. Afterwards, the incidence of neonatal hypothermia and short-term outcomes, such as intraventricular hemorrhage (IVH), acidosis, and shock requiring inotropic agents, in each group were compared.

### Results

A total of 95 VLBW infants were enrolled in the study, including 37 pre-intervention, and 58 post-intervention cases. The incidence of hypothermia in preterm infants decreased significantly upon the implementation of our prevention bundle, both in the delivery room (from

**Funding:** The authors received no specific funding for this work.

**Competing interests:** The authors have declared that no competing interests exist.

45.9% to 8.6%) and on admission (59.5% to 15.5%). In addition, the short-term outcomes of VLBW infants improved significantly, especially with the decreased incidence of IVH (from 21.6% to 5.2%, P = 0.015).

## Conclusions

Our standardized prevention bundle for preventing hypothermia in VLBW infants is effective and decreased the IVH rate in VLBW infants. We strongly believe that this prevention bundle is a simple, low-cost, replicable, and effective tool that hospitals can adopt to improve VLBW infant outcomes.

## Introduction

Cold stress may help initiate neonatal respiration. It may also play a protective role in the central nervous system of a newborn with perinatal asphyxia [1]. However, prolonged exposure to cold has been associated with many complications, including hypoglycemia [2], respiratory distress [3], bronchopulmonary dysplasia [4], necrotizing enterocolitis (NEC) [5], metabolic acidosis [6], intraventricular hemorrhage (IVH) [3, 7], late-onset sepsis [3, 8], and even death [9, 10].

The World Health Organization (WHO) defines neonatal hypothermia as a temperature of less than 36.5˚C [11]. The association between hypothermia on admission and neonatal death has been well-documented. Upon admission, every 1˚C decrease in temperature of a newborn with low birth weight (LBW) is associated with a 28% increase in the incidence of in-hospital mortality and an 11% increase in the incidence of late-onset sepsis [8]. Both early (1–6 days of age) and late (7–28 days of age) mortality are also higher among the group of neonates with an admission temperature of <35˚C [12]. Conversely, every 1˚C increase in admission temperature is associated with a 15%–80% decrease in mortality [12, 13]. In addition, the combined mortality/morbidity rate in preterm infants born prior to 33 weeks of gestation was the lowest among the group of neonates with admission temperatures ranging from 36.5˚C to 37.2˚C [14].

During the first 12 hours of life, the ability of an infant to initiate physiological mechanisms (e.g., shivering) for maintaining his/her body temperature is limited. Hence, newborns struggle to maintain their body temperature. Therefore, they rely on external sources of warmth [15, 16]. If adequate warmth is not provided immediately after birth, both the core and skin temperatures of a full-term baby decrease gradually at a rate of 0.1˚C/min and 0.3˚C /min, respectively [17].

Other risk factors that increase the difficulty experienced by newborns in maintaining a warm body temperature include a relatively high body surface area to body weight ratio, a thinner layer of subcutaneous fat, fewer glycogen stores, immature skin barriers, and a low room temperature in the delivery or operating room [18]. Preterm infants experience more heat loss through evaporation than term infants [18]. Every 1 mL of water that is lost via evaporation is associated with 560 calories that are also lost [19]. Among preterm infants born with LBW [20], intrauterine growth retardation [21], perinatal asphyxia, congenital anomaly, or central nervous injury [15], the rate of admission hypothermia is especially high.

Despite the existing guidelines for the prevention of hypothermia, maintaining sufficient warmth in preterm babies, especially during resuscitation, remains challenging. This challenge exists regardless of the regional climate the babies are born in and the availability of the

resources around them at birth. Statistics presented by the Vermont Oxford Network in 2016 revealed that, although the incidence of hypothermia on admission decreased significantly from 52.6% to 38.2% between 2009 and 2016, the incidence was still relatively high [22]. The same study also found that the rate of admission hypothermia with a temperature below 36˚C, which is defined as moderate-to-severe hypothermia, was 14.8%, with the highest prevalence occurring in the group with the lowest birth BW (BBW) and gestational age (GA) [22]. In the UK, the National Neonatal Audit Programme considers preterm hypothermia as a problem that requires more attention and one that has much room for improvement—the rates of admission hypothermia among preterm infants were as high as 28% and 25% in 2015 and 2016, respectively [23, 24]. Therefore, developing a feasible strategy to prevent hypothermia in very low BW (VLBW) infants is crucial and necessary. Herein, we aimed to design a standard prevention bundle, which is a simple set of evidence-based practices that improve the reliability of their delivery and improve patient outcomes when implemented collectively [25], to prevent hypothermia after birth, as well as to investigate efficacy of the bundle and short-term outcomes of VLBW infants.

## Materials and methods

### Study design

This study was conducted at a single referral level III neonatal intensive care unit (NICU) in the National Taiwan University Children Hospital. All infants born between February 2017 and July 2018 with a BW of <1,500 g or GA of <31 weeks were included. We anticipated collecting approximately 100 cases to achieve a 40% reduction in the admission hypothermia rate considering a previous study [26], the duration of this quality improvement (QI) project provided by the government, and the average birth numbers of VLBW infants per month in our hospital.

The study was classified according to two time periods: pre-intervention (February to September 2017) and post-intervention (October 2017 to July 2018). We collected the patients' data, which included the delivery room (DR) temperature, body temperature, and incidence of certain complications. Then, we formed a multidisciplinary QI team that was comprised of neonatologists, obstetricians, and nurses to determine the potential factors causing neonatal hypothermia soon after birth. A fishbone diagram (Fig 1) and Pareto chart (Fig 2) were then used to clarify the leading factors, and solutions targeting the said causes were proposed. Next,

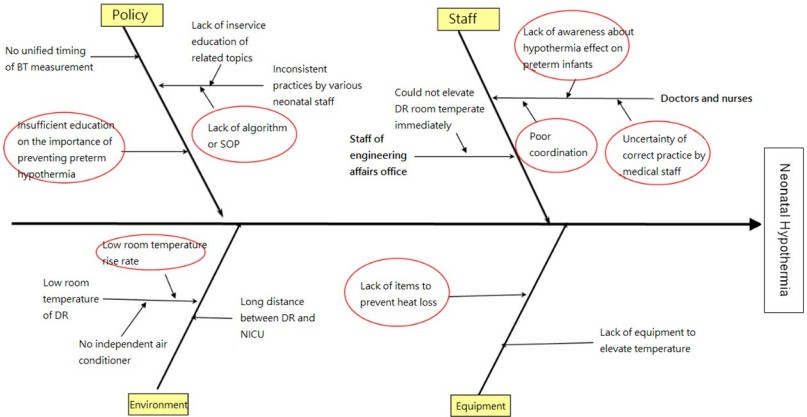

**Fig 1. Fishbone diagram for potential factors of neonatal hypothermia.**

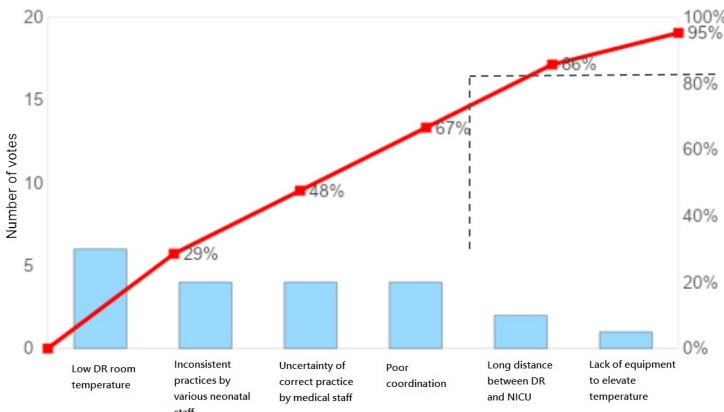

**Fig 2. Pareto chart for leading factors of neonatal hypothermia.**

we developed a prototype of the hypothermia prevention bundle and continued to refine it until it was a sound and reliable tool. The final version of the prevention bundle, which serves as the intervention used in this study, was established and presented as an algorithm by the end of September 2017 to assure standardization. During the post-intervention period, we continued to collect data until the completion of the study in July 2018.

The study was approved by the Institutional Review Board of the National Taiwan University Hospital (IRB number: 202107151RINB). The review board waived the requirement of obtaining informed consent since all data were collected retrospectively after the QI project following national policy.

## Primary outcomes

Neonatal hypothermia is defined as a rectal temperature below 36.5˚C [27]. We collected basic clinical characteristics of VLBW infants, DR temperature, the incidence of hypothermia in the DR (based on the rectal temperature before leaving the delivery room), and rate of hypothermia on admission (based on the rectal temperature right after arriving at the NICU).

## Secondary outcomes

Several complications that contribute to the high mortality and morbidity of neonatal hypothermia, including IVH, acidosis, hypotension, and respiratory distress, were noted in this study. Other short-term complications with relatively low prevalence, such as confirmed NEC or late-onset sepsis, were not investigated considering our study period and sample size. IVH of any grade, revealed by routine cranial ultrasound, was included; hypotension was defined as mean blood pressure (mmHg) below the post-menstrual age (weeks) and requiring inotropic agents; acidosis was defined as an initial blood gas pH of <7.2.

## Statistical analysis

The Chi-square test was used for the categorical data, while the independent samples t-test was used for continuous data. IBM SPSS statistics version 28 (IBM Corp., Armonk, N.Y., USA) was used for all statistical analyses. Statistical significance was defined as a P value <0.05.

## Results

### Standardization of the prevention bundle for neonatal hypothermia

During the pre-intervention period, three leading causes of hypothermia for VLBW infants were identified. Our solutions to these causes are listed below.

**Low DR temperature.**   The average recorded DR temperature was 19˚C, which was considered significantly low when caring for VLBW infants. The higher the DR temperature, the easier it was to warm preterm babies. However, the following factors were also considered while regulating the DR temperature: the comfort of the obstetrician and the challenge of avoiding sweating to prevent the spread of infection during delivery. Thus, a higher DR temperature might not always be better, especially if there are other means to keep the baby warm. After negotiating with the obstetricians, we decided to increase the DR temperature from 19˚C to 21˚C as soon as we were notified about a pending preterm delivery. Additionally, we also set up a portable radiant warmer and placed at least one heat lamp beside it in the DR.

**Inconsistent practices by various neonatal staff.**   We designed a standard operating procedure for the care bundle that aimed to prevent preterm hypothermia (Fig 3). This involved using a checklist to ensure that every step was completed, which also made us more confident that all tasks would be performed correctly and efficiently. Additionally, all the necessary items needed to prevent preterm hypothermia were placed in a spare bag and included a cotton cap, a plastic wrap, a thermometer, and a circuit for nasal continuous positive airway pressure.

**Uncertainty of correct practice by medical staff.**   We created an instructional video regarding correct medical practices for the medical staff that was uploaded to YouTube (https://youtu.be/vLjqwh8PgaA). Every resident on rotation was tasked to watch the video completely before working in the NICU. Additionally, we used team resource management strategies to ensure that the new staff members were learning the correct medical practices. We used a checklist to assure the validity and reliability of the above practice (Table 1). Additionally, the completion rate was above 90%.

During the post-intervention period, we continued to recruit participants and collect data. By the end of the study, a total of 95 cases were successfully recruited, including 37 cases during the pre-intervention period (between February 2017 and September 2017) and 58 cases during the post-intervention period (between October 2017 and July 2018).

The demographic data and clinical characteristics of the two groups are summarized in Table 2. No significant differences between the two groups were observed in the following parameters: GA, BBW, rate of small for gestational age (SGA), sex, mode of delivery, rate of antenatal steroid use, rate of premature rupture of membrane, and 1-min Apgar score.

The average DR temperature increased from 20.56˚C ± 1.40˚C to 21.45˚C ± 1.30˚C. The average body temperatures in the DR and upon arriving at the NICU were significantly higher in the post-intervention group. The incidence of hypothermia in the DR decreased from 45.9% to 8.6%, and the rate of admission hypothermia decreased from 59.5% to 15.5%. We also found that the rate of IVH significantly decreased in the post-intervention group (21.6% vs. 5.2%, P = 0.015), while other short-term outcomes, such as the 5-min Apgar score (10.8% vs. 5.2%, P = 0.305), initial acidosis (16.7% vs. 13.7%, P = 0.719), and shock that required inotropic agents (18.9% vs. 15.5%, P = 0.666) showed decreasing trends (Table 3).

## Discussion and conclusions

After applying our care bundle approach, the incidence of hypothermia both in the DR and on admission decreased significantly. In our hospital, the hypothermia rates prior to the use of the intervention were 45.9% in the DR and 59.5% on admission, which are similar to those found

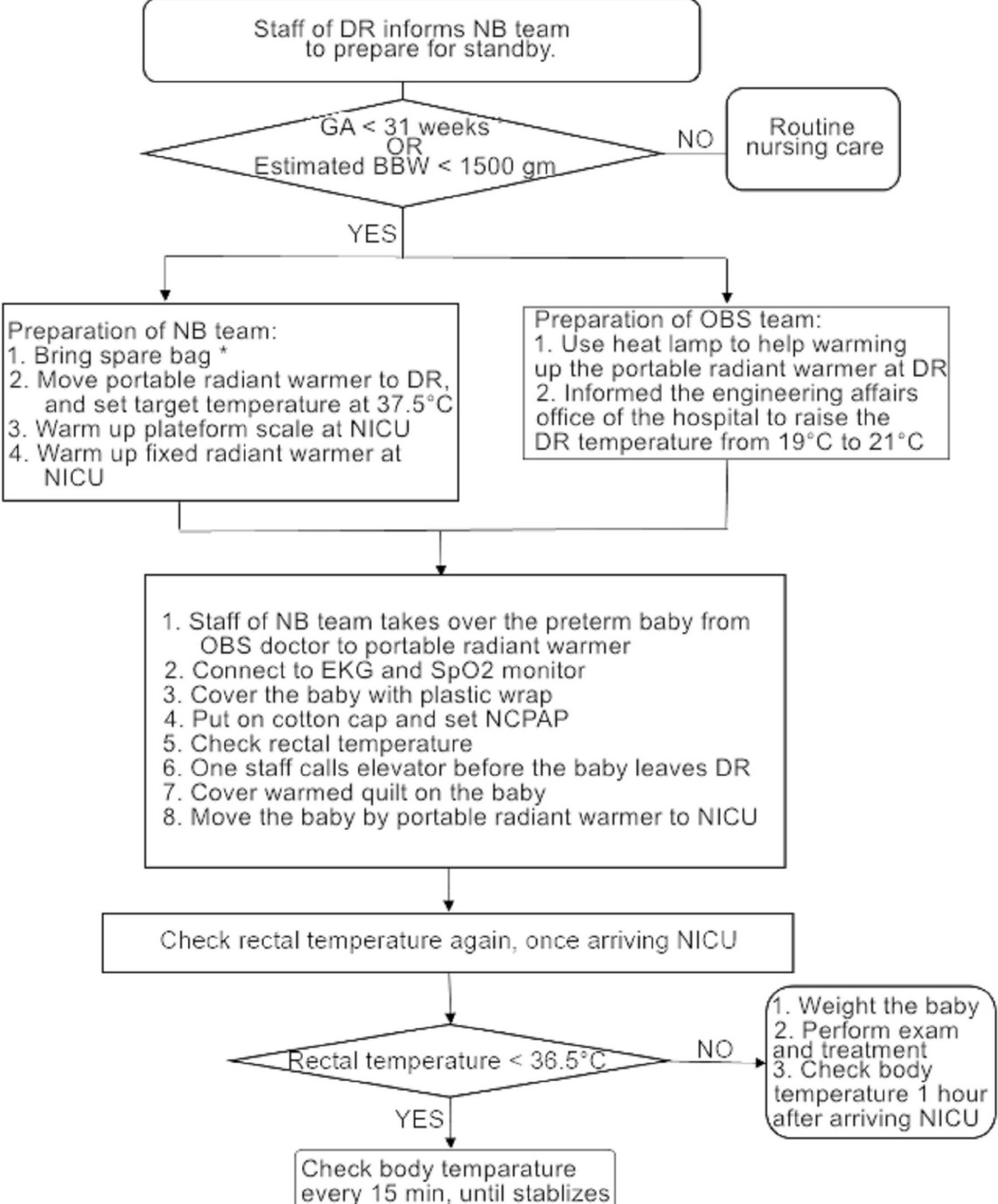

**Fig 3. Algorithm of care bundle approach to prevent preterm hypothermia.** DR, delivery room; NB, newborn; GA, gestational age; BBW, birth body weight; NICU, neonatal intensive care unit; OBS, obstetrics; NCPAP, nasal continuous positive airway pressure. [†]Contents of the spare bag include a cotton cap, plastic wrap, thermometer, and a circuit for nasal continuous positive airway pressure.

in previous studies [3, 13, 27–32]. However, the hypothermia rates after intervention were 8.6% in the DR and 15.5% on admission, which were both lower than those of other hospitals [27–30, 33–37]. The discrepancy between these findings may be attributed to the fact that previous studies focused on single variables (e.g., to wrap or not, different wrapping materials,

**Table 1. Checklist for the prevention bundle of neonatal hypothermia.**

| Assessment Items | Assessment Contents |
|---|---|
| Object | 1. BBW of <1,500 g or GA of <31 weeks |
| Prenatal Preparation | 2–1. Preheat radiant warmer and platform scale in NICU 30 min before delivery |
| | 2–2. Preheat portable radiant warmer in DR 30 min before delivery |
| | 2–3. Put a dry, warm towel on the portable radiant warmer in DR |
| | 2–4. Place cotton cap and plastic wrap on the towel |
| | 2–5. Confirm the room temperature of the DR and place at least one heat lamp beside the portable radiant warmer |
| Postnatal Managements in DR | 3–1. Use plastic wrap to cover the patient's whole body except the head |
| | 3–2. Put on cotton cap |
| | 3–3. Record rectal temperature before leaving the DR |
| | 3–4. Keep above measures in position |
| | 3–5. Check the function of the portable radiant warmer |
| | 3–6. Ensure the patient is being covered by the plastic wrap and cotton cap adequately |
| | 3–7. Cover the patient with a warm blanket |
| Postnatal Managements in NICU | 4–1. Record the rectal temperature on the portable radiant warmer once entering the NICU |
| | 4–2. Record the BW and move to preheated radiant warmer in NICU; recheck body temperature after 1 hour |
| | 4–3. Postpone unnecessary examinations until the patient's body temperature stabilizes |
| | 4–4. Record the body temperature every 15 mins to check if hyperthermia (>38°C) or hypothermia (<36.5°C) occurs |
| | 4–5. Use an external heat source to maintain the body temperature, including a radiant warmer, warm blanket, warm water bag (40–42°C), and increasing the temperature of the incubator or ventilator |
| | 4–6. Remove the external heat source, as mentioned above, if hyperthermia occurs |
| | 4–7. Consider infection if hyperthermia or hypothermia persists for more than 1 hour |

**Table 2. Comparison of infant variables.**

| Parameter | | Pre-/intra-intervention | Post-intervention | P value |
|---|---|---|---|---|
| | | N = 37 | N = 58 | |
| Gestational age, mean (SD) | | 30.0 (3.0) | 28.9 (3.2) | 0.115 |
| Birth weight, mean (SD) | | 1,103 (299) | 1,052 (311) | 0.428 |
| Cesarean section | Yes | 31 (83.8) | 48 (82.8) | 0.896 |
| | No | 6 (16.2) | 10 (17.2) | |
| Sex | Male | 14 (37.8) | 31 (53.4) | 0.137 |
| | Female | 23 (62.2) | 27 (46.6) | |
| SGA | Yes | 21 (56.8) | 25 (43.1) | 0.194 |
| | No | 16 (43.2) | 33 (56.9) | |
| Antenatal steroid | Yes | 30 (81.1) | 54 (93.1) | 0.074 |
| | No | 7 (18.9) | 4 (6.9) | |
| PPROM | Yes | 12 (32.4) | 29 (50.0) | 0.092 |
| | No | 25 (67.6) | 29 (50.0) | |
| 1-min Apgar <7 | Yes | 27 (73.0) | 42 (72.4) | 0.952 |
| | No | 10 (27.0) | 16 (27.6) | |

Abbreviations: SGA, small for gestational age; PPROM, preterm-premature rupture of membrane

**Table 3. Primary and secondary outcomes between the pre-/intra-intervention and post-intervention groups.**

| Parameter | | Pre-intervention | Post-intervention | P value |
|---|---|---|---|---|
| | | N = 37 | N = 58 | |
| DRT, mean (SD) | | 20.56 (1.40) | 21.45 (1.30) | 0.002 |
| DRT ≥21°C | | 12 (33.3) | 39 (68.4) | <0.001 |
| BT1, mean (SD) [†] | | 36.5 (0.35) | 36.8 (0.40) | <0.001 |
| BT2, mean (SD) | | 36.4 (0.38) | 36.7 (0.39) | <0.001 |
| BT1 <36.5°C [†] | | 17 (45.9) | 5 (8.6) | <0.001 |
| BT2 <36.5°C | | 22 (59.5) | 9 (15.5) | <0.001 |
| BT1 ≥38°C [†] | | 0 (0) | 1 (1.8) | 0.418 |
| BT2 ≥38°C | | 0 (0) | 1 (1.7) | 0.422 |
| Death | Yes | 1 (2.7) | 0 (0) | 0.208 |
| | No | 36 (97.3) | 58 (100) | |
| Survanta | Yes | 4 (10.8) | 8 (13.8) | 0.670 |
| | No | 33 (89.2) | 50 (86.2) | |
| IVH | Yes | 8 (21.6) | 3 (5.2) | 0.015 |
| | No | 29 (78.4) | 55 (94.8) | |
| Inotropic agent use | Yes | 7 (18.9) | 9 (15.5) | 0.666 |
| | No | 30 (81.1) | 49 (84.5) | |
| pH <7.2 [‡] | Yes | 5 (16.7) | 7 (13.7) | 0.719 |
| | No | 25 (83.3) | 44 (86.3) | |
| 5-min Apgar <7 | Yes | 4 (10.8) | 3 (5.2) | 0.305 |
| | No | 33 (89.2) | 55 (94.8) | |

Abbreviations: DRT, delivery room temperature; BT1, body temperature before leaving the delivery room; BT2, body temperature at the NICU; IVH, intraventricular hemorrhage

[†]Only 1 patient was missing this variable

[‡]A total of 14 patients were missing this variable

different delivery room temperatures), whereas our study simultaneously looked at various relevant parameters. A recent meta-analysis also suggests utilizing a combination of interventions for the most vulnerable neonates [38].

As suggested by the WHO, an adequate DR temperature between 25°C and 28°C should be maintained as this reduces the incidence of neonatal hypothermia [11]. Previous studies have also demonstrated that the rate of hypothermia on admission was significantly lower if the DR temperature was maintained between 23.3°C and 25°C [33, 39]. However, we should also consider the importance of preventing the staff from sweating excessively to maintain an aseptic field during delivery. As advised by our hospital's obstetricians, we increased the DR temperature from 19°C to 21°C prior to the delivery of all VLBW infants. This change, coupled with the care bundle approach, resulted in a similar, or perhaps even better, effect in the prevention of preterm hypothermia.

Although previous studies have described the relationship between hypothermia and various complications, most interventional studies either lack a study design that is able to compare the complications between the group with a decreased incidence of hypothermia and the control group or fail to demonstrate a significant difference between groups. Our study has demonstrated that our care bundle approach did not only decrease the incidence of neonatal hypothermia in VLBW infants but also reduced the rate of IVH. IVH is a significant cause of brain injury in newborns and occurs most frequently in VLBW infants and/or very preterm

infants (GA <32 weeks). The overall risk of IVH in very preterm infants is 20.2% [40], and it is associated with admission hypothermia [3, 7]. The incidence rate of IVH in our hospital was 21.6%, which is comparable with that of previous studies; however, this decreased significantly (5.2%) after the implementation of our prevention care bundle.

In conclusion, our standardized prevention bundle for preventing hypothermia in VLBW infants is effective. Additionally, decreasing the incidence of hypothermia in VLBW infants led to a subsequent decrease in the incidence of short-term complications such as IVH. We believe that this low-cost prevention bundle is a simple, replicable, and robust tool for decreasing the incidence of hypothermia and associated complications among VLBW infants in hospitals around the world.

## Supporting information

**S1 Table. Raw data of participants in the study.**
(XLSX)

**S1 Checklist. Reporting checklist for cohort study.**
(DOCX)

## Acknowledgments

We acknowledge the help provided by our nurses and residents, especially with data collection. We would also like to thank the Center for Quality Management of National Taiwan University Hospital for their unending support in developing this neonatal hypothermia prevention bundle.

## Author Contributions

**Conceptualization:** Yi-Hsuan Lu, Yin-Ting Chen, Yin Ling Tan, Yu-Lien Chen, Hung-Chieh Chou, Chien-Yi Chen, Ting-An Yen, Po-Nien Tsao.

**Data curation:** Wei-Tse Chiu, Yi-Hsuan Lu, Yin-Ting Chen, Yu-Lien Chen, Po-Nien Tsao.

**Formal analysis:** Wei-Tse Chiu, Yi-Hsuan Lu, Yin-Ting Chen, Po-Nien Tsao.

**Investigation:** Wei-Tse Chiu, Yi-Hsuan Lu, Yin-Ting Chen, Yin Ling Tan, Yi-Chieh Lin, Po-Nien Tsao.

**Methodology:** Po-Nien Tsao.

**Project administration:** Wei-Tse Chiu, Yi-Hsuan Lu, Yin-Ting Chen, Yin Ling Tan, Yi-Chieh Lin, Yu-Lien Chen, Po-Nien Tsao.

**Writing – original draft:** Wei-Tse Chiu, Po-Nien Tsao.

**Writing – review & editing:** Hung-Chieh Chou, Chien-Yi Chen, Ting-An Yen.

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
