## [Decision Letter · Decision Letter 0]

8 Mar 2022

PONE-D-21-38625Neonatal hypothermia improvement following implementation of a feasible prevention bundlePLOS ONE

Dear Dr. Tsao,

Thank you for submitting your manuscript to PLOS ONE. After careful consideration, we feel that it has merit but does not fully meet PLOS ONE’s publication criteria as it currently stands. Therefore, we invite you to submit a revised version of the manuscript that addresses the points raised during the review process.

We look forward to receiving your revised manuscript.

Kind regards,

Linglin Xie

Academic Editor

PLOS ONE

Journal Requirements:

Reviewers' comments:

Reviewer's Responses to Questions

**Comments to the Author**

1. Is the manuscript technically sound, and do the data support the conclusions?

Reviewer #1: Partly

2. Has the statistical analysis been performed appropriately and rigorously? 

Reviewer #1: No

3. Have the authors made all data underlying the findings in their manuscript fully available?

Reviewer #1: Yes

4. Is the manuscript presented in an intelligible fashion and written in standard English?

Reviewer #1: No

5. Review Comments to the Author

Reviewer #1: Title

The title is not parallel to the objectives of the study.

Abstract

Introduction: please describe about justification of the study?

Methods: the study design and setting, sample size, data collection method, source population, eligibility criteria, data analysis and Etc. are not clearly described in the method section of the abstract.

Discussion/conclusion: please remove the word discussion. And what is the recommendation based on the findings of the study?

Introduction

Define hypothermia here. Please show the gap in the introduction section of the manuscript.

Methods and materials

What is the word “general”? Please re-write the methods and materials section separately by incorporating the sub-element study period, study design, eligibility criteria, data collection procedure, sample size and sampling procedure, operational definition ….

Please operationalized your outcome variables and prevention bundle?

What does it mean the word feasible and effective? Please also operationalized this word?

In general, please re-write it clearly?

Results

Why the pre-intervention and post-intervention groups are different in number?

Discussion

Please re-write based on the finding of the study

6. PLOS authors have the option to publish the peer review history of their article (what does this mean?). If published, this will include your full peer review and any attached files.

Reviewer #1: No

---

## [Author Response · Author response to Decision Letter 0]

3 May 2022

Response to editor

1. Please ensure that your manuscript meets PLOS ONE's style requirements, including those for file naming. The PLOS ONE style templates can be found at https://journals.plos.org/plosone/s/file?id=wjVg/PLOSOne_formatting_sample_main_body.pdf, and https://journals.plos.org/plosone/s/file?id=ba62/PLOSOne_formatting_sample_title_authors_affiliations.pdf

Response: We thank the reviewer for this important reminder. We have reviewed our manuscript thoroughly and ensured that the formatting is in order.

Response: We thank this timely reminder. We have uploaded our study’s data set under the Supporting Information files section.

Response to reviewers

Reviewer #1:

- Title

The title is not parallel to the objectives of the study.

Response: We thank the reviewer for this pertinent comment. We have revised our title to better reflect our study objectives, as suggested by the reviewer. Our new title is: “Reducing intraventricular hemorrhage following the implementation of a prevention bundle for neonatal hypothermia” and this has been included in our revised manuscript.

- Abstract

- Introduction: please describe about justification of the study?

Response: We thank the reviewer for this timely reminder. We have revised our Introduction section to reflect the justification and objectives of the study, as suggested by the reviewer. Please refer to page 3, lines 31-32 in the manuscript (mark in red). We have established that hypothermia immediately after birth is a common, but difficult to prevent challenge among very low birth weight (VLBW) infants worldwide. For this reason, we have designed a protocol that acts as a prevention bundle to improve infant outcomes.

- Methods: the study design and setting, sample size, data collection method, source population, eligibility criteria, data analysis and Etc. are not clearly described in the method section of the abstract.

Response: We thank the reviewer for this pertinent comment. We have revised our Abstract session, as suggested by the reviewer. Please refer to page 3, lines 36-45. We have rewritten this part of our Abstract to ensure comprehension and readability.

- Discussion/conclusion: please remove the word discussion. And what is the recommendation based on the findings of the study?

Response: We thank the reviewer for this important comment. We have revised the section header, as suggested by the reviewer. Please refer to page 4, lines 54-57. We believe that this protocol is an effective, simple, low-cost, and replicable tool for most of the hospitals. Readers may refer to our algorithm to improve the outcomes of preterm hypothermia in their hospital. 

- Introduction

Define hypothermia here. Please show the gap in the introduction section of the manuscript.

Response: We thank the reviewer for this comment. We have revised our Introduction section to highlight the definition of hypothermia, as well as provide more background and key literature related to our study. Please refer to page 4, lines 66-67 and page 6 lines 95-106. The World Health Organization (WHO) defines neonatal hypothermia as a temperature of less than 36.5 °C. We have also highlighted the high incidence of preterm hypothermia in developed countries.

- Methods and materials

What is the word “general”? Please re-write the methods and materials section separately by incorporating the sub-element study period, study design, eligibility criteria, data collection procedure, sample size and sampling procedure, operational definition ….

Please operationalized your outcome variables and prevention bundle? 

What does it mean the word feasible and effective? Please also operationalized this word?

In general, please re-write it clearly?

Response: We thank the reviewer for these pertinent comments. We have revised our manuscript to reflect these improvements, as suggested by the reviewer. Please refer to page 7, lines 115-135. We have rewritten the study design and attached the algorithm for our prevention bundle (Fig 1) to ensure reader comprehension and readability.

- Results

Why the pre-intervention and post-intervention groups are different in number? 

Response: We thank the reviewer for this question. This study was retrospective in nature and based on a previous project of quality improvement following national policy. The beginning and the ending of enrollment were both based on national policy, so the numbers of participants in each group (grouped based on their recruitment periods), could not be controlled. Additionally, the effect of prevention bundle in improving outcomes might fade gradually, because there would not be repeated intensive training for medical staff once the standardized SOP had been established, and their skill to execute the prevention bundle correctly as time goes on might be a reasonable concern. Thus, we tried to collect as many infants for the post-intervention group as we could, to highlight the effectiveness and sustainability of our prevention bundle. 

- Discussion

Please re-write based on the finding of the study

Response: We thank the reviewer for this suggestion. We have revised our Discussion section to reflect the findings of our study, as suggested by the reviewer. Please refer to page 14 lines 221-227, 233-236, and 240-253 in the manuscript (mark in red). We also compared our study to a recent meta-analysis, cited more references and emphasized the clinical significance of reducing the IVH rate among infants after using our prevention bundle.

---

## [Decision Letter · Decision Letter 1]

18 May 2022

PONE-D-21-38625R1Reducing intraventricular hemorrhage following the implementation of a prevention bundle for neonatal hypothermiaPLOS ONE

Dear Dr. Tsao,

Thank you for submitting your manuscript to PLOS ONE. After careful consideration, we feel that it has merit but does not fully meet PLOS ONE’s publication criteria as it currently stands. Therefore, we invite you to submit a revised version of the manuscript that addresses the points raised during the review process.

We look forward to receiving your revised manuscript.

Kind regards,

Linglin Xie

Academic Editor

PLOS ONE

Journal Requirements:

Reviewers' comments:

Reviewer's Responses to Questions

**Comments to the Author**

1. If the authors have adequately addressed your comments raised in a previous round of review and you feel that this manuscript is now acceptable for publication, you may indicate that here to bypass the “Comments to the Author” section, enter your conflict of interest statement in the “Confidential to Editor” section, and submit your "Accept" recommendation.

Reviewer #1: (No Response)

2. Is the manuscript technically sound, and do the data support the conclusions?

Reviewer #1: Partly

3. Has the statistical analysis been performed appropriately and rigorously? 

Reviewer #1: Yes

4. Have the authors made all data underlying the findings in their manuscript fully available?

Reviewer #1: Yes

5. Is the manuscript presented in an intelligible fashion and written in standard English?

Reviewer #1: Yes

6. Review Comments to the Author

Reviewer #1: Abstract

General comment: please correct the major editorial problem like alignment……..

Introduction: In line 32 of the abstract section, there is a word…..difficult to prevent…... However, your study intended to investigate about the efficacy of the prevention bundles. The two things are contradicting each other’s.

Methods: what type of retrospective study does your study used? Please say something about sample size calculations in the methodology section of the manuscript. What are the study units? Is it VLBW, preterm neonates or both?

Result: Why the study intended to investigate only the stated short-term outcomes of hypothermia? Why other outcomes of hypothermia are not investigated yet?

Methods

Study design: please write introductory sentence about prevention bundle? When you say the prevention protocols are reliable and sound? How you are going to make the protocol standardized?

Outcomes: how you are going to measure hypotension and IVH?

Please re-write the methodology by considering what things has to be incorporated in each component of methods and materials? Why you classified the study group in to three? Why do not you classified as pre and post intervention group? Why the study groups differ in number?

Results

Low DR temperature: what is the parameter to increase the DR temperature from 19 to 21? Sentence in page 9, line 170-174 is not clear for reader, please write clearly? Institutional video: how do you assure the validity and reliability of the video? “….whereas the incidence of hyperthermia did not increase significantly between the two groups.” Delete this phrase?

Discussion and conclusion

WHO recommendation of DR temperature is 25 to 28. However, you upgrade DR temperature in your study setting from 19 to 21. Why?

The association of prevention bundle and mortality are not investigated yet in the study and you concluded that the prevention bundles are effective to reduce preterm mortality. However, the conclusions and interpretation has to be based on the finding of the study.

7. PLOS authors have the option to publish the peer review history of their article (what does this mean?). If published, this will include your full peer review and any attached files.

Reviewer #1: No

---

## [Author Response · Author response to Decision Letter 1]

5 Aug 2022

Response to editor 

Response: We have reviewed and revised our reference list. (lines 333-345) 

Response to reviewer

1. General comment: please correct the major editorial problem like alignment……..

Response: The format of the manuscript has been reviewed and revised by the authors and English-editoring company. (editing certificate is uploaded as supporting information)

2. Introduction: In line 32 of the abstract section, there is a word…..difficult to prevent…... However, your study intended to investigate about the efficacy of the prevention bundles. The two things are contradicting each other’s.

Response: We have substituted the phrase “difficult to prevent” with “common even in countries rich in medical resources.” (line 32)

3. Methods: what type of retrospective study does your study used? Please say something about sample size calculations in the methodology section of the manuscript. What are the study units? Is it VLBW, preterm neonates or both?

Response: 

(1) We have revised the Methods section of the manuscript. The study was a quality-improvement project. (line 36)

(2) We anticipated enrolling 100 cases to achieve a 40% reduction in the admission hypothermia rate while considering a previous study, the duration of this QI project provided by the government, and the average number of VLBW infant births per month in our hospital. (lines 114-117)

(3) We recruited every newborn with a BBW <1,500 gm or GA <31 weeks during the study period. (lines 113-114)

4. Result: Why the study intended to investigate only the stated short-term outcomes of hypothermia? Why other outcomes of hypothermia are not investigated yet?

Response: Other short-term complications with relatively low prevalence, such as necrotizing enterocolitis or late-onset sepsis, were not investigated considering our study period and sample size. (lines 147–148)

5. [Methods] Study design: please write introductory sentence about prevention bundle? When you say the prevention protocols are reliable and sound? How you are going to make the protocol standardized?

Response: 

(1) We aimed to design a standard hypothermia prevention bundle, which is a simple set of evidence-based practices that improve the reliability of their delivery and improve patient outcomes when implemented collectively, to prevent hypothermia after birth and investigate the bundle’s efficacy and short-term outcomes of VLBW infants. (lines 105-108)

(2) The final version of the prevention bundle, which serves as the intervention used in this study, was established and presented as an algorithm by the end of September 2017 to assure standardization. (lines 126-128)

6. Outcomes: how you are going to measure hypotension and IVH?

Response: IVH of any grade, revealed by routine cranial ultrasound, was included; hypotension was defined as mean blood pressure (mmHg) below the post-menstrual age (weeks) and requiring inotropic agents; acidosis was defined as an initial blood gas pH of <7.2. (lines 147–151)

7. Please re-write the methodology by considering what things has to be incorporated in each component of methods and materials? Why you classified the study group in to three? Why do not you classified as pre and post intervention group? Why the study groups differ in number? 

Response: We have revised and rephrased the Methods section as suggested, including combining the pre- and intra-intervention groups into one group, the thinking process by which the algorithm was formed, and the reason for recruiting certain case numbers. 

8. Low DR temperature: what is the parameter to increase the DR temperature from 19 to 21? Sentence in page 9, line 170-174 is not clear for reader, please write clearly? Institutional video: how do you assure the validity and reliability of the video? “….whereas the incidence of hyperthermia did not increase significantly between the two groups.” Delete this phrase?

Response:

(1) The higher the DR temperature, the easier it was to warm preterm babies. However, the following factors were also considered while regulating the DR temperature: the comfort of the obstetrician and the challenge of avoiding sweating to prevent the spread of infection during delivery. Thus, a higher DR temperature might not always be better, especially if there are other means to keep the baby warm. After negotiating with the obstetricians, we decided to increase the DR temperature from 19°C to 21°C as soon as we were notified about a pending preterm delivery. (lines 165–171)

(2) Line 180-185 is caption and legend of Figure 3. 

(3) We used a checklist to assure the validity and reliability of the video. The completion rate was above 90%. (lines 191-192)

(4) We have deleted the phrase, as suggested.

9. WHO recommendation of DR temperature is 25 to 28. However, you upgrade DR temperature in your study setting from 19 to 21. Why?

Response: The higher the DR temperature, the easier it was to warm preterm babies. However, the following factors were also considered while regulating the DR temperature: the comfort of the obstetrician and the challenge of avoiding sweating to prevent the spread of infection during delivery. Thus, a higher DR temperature might not always be better, especially if there are other means to keep the baby warm. After negotiating with the obstetricians, we decided to increase the DR temperature from 19°C to 21°C as soon as we were notified about a pending preterm delivery. (lines 165–171)

10. The association of prevention bundle and mortality are not investigated yet in the study and you concluded that the prevention bundles are effective to reduce preterm mortality. However, the conclusions and interpretation has to be based on the finding of the study.

Response: We have revised our wording as suggested. We believe that this low-cost prevention bundle is a simple, replicable, and robust tool for decreasing the incidence of hypothermia and associated complications among VLBW infants in hospitals around the world. (lines 259–261)

---

## [Editor Report · Decision Letter 2]

19 Aug 2022

Reducing intraventricular hemorrhage following the implementation of a prevention bundle for neonatal hypothermia

PONE-D-21-38625R2

Dear Dr. Tsao,

We’re pleased to inform you that your manuscript has been judged scientifically suitable for publication and will be formally accepted for publication once it meets all outstanding technical requirements.

Kind regards,

Linglin Xie

Academic Editor

PLOS ONE
---

## [Editor Report · Acceptance letter]

26 Aug 2022

PONE-D-21-38625R2 

Reducing intraventricular hemorrhage following the implementation of a prevention bundle for neonatal hypothermia 

Dear Dr. Tsao:

I'm pleased to inform you that your manuscript has been deemed suitable for publication in PLOS ONE. Congratulations! Your manuscript is now with our production department. 

Kind regards, 

on behalf of

Dr. Linglin Xie 

Academic Editor

PLOS ONE